# Spatial control over catalyst positioning on biodegradable polymeric nanomotors

B. Jelle Toebes[1,2], F. Cao [1,2] & Daniela A. Wilson [1*]

Scientists over the world are inspired by biological nanomotors and try to mimic these complex structures. In recent years multiple nanomotors have been created for various fields, such as biomedical applications or environmental remediation, which require a different design both in terms of size and shape, as well as material properties. So far, only relatively simple designs for synthetic nanomotors have been reported. Herein, we report an approach to create biodegradable polymeric nanomotors with a multivalent design. PEG-PDLLA (poly (ethylene glycol)-b-poly($D,L$-lactide)) stomatocytes with azide handles were created that were selectively reduced on the outside surface by TCEP (tris(2-carboxyethyl)phosphine) functionalized beads. Thereby, two different functional handles were created, both on the inner and outer surface of the stomatocytes, providing spatial control for catalyst positioning. Enzymes were coupled on the inside of the stomatocyte to induce motion in the presence of fuel, while fluorophores and other molecules can be attached on the outside.

[1] Systems Chemistry Department, Institute for Molecules and Materials, Radboud University Nijmegen, Nijmegen, The Netherlands. [2]These authors contributed equally: B. Jelle Toebes, F. Cao. *email: d.wilson@science.ru.nl

Self-propelled nanomotors are small machines that are powered by catalytic reactions in fluids[1,2]. The design and demands of these motors depend strongly on their tasks, e.g., for biomedical applications, the motors should be made from soft biocompatible and/or biodegradable materials and should be adaptive like biological motors in nature[3–7]. An adaptive system requires a multivalent design that combines mobility together with other functionalities, like sensing[8], recognition[9], and/or other active components[10,11], to ensure that the motor can respond to changes in the environment. So far, the design of existing nanomotors is relatively simple, limiting their application possibilities. Implementing multiple functionalities in one system is still challenging and laborious, yet should be easy and straightforward, especially when going for more complex settings, i.e. mimicking life-like materials. A crucial factor in the design of nanomotors is positional control of the catalyst, which is difficult to achieve at the nanoscale[12,13]. If it is possible to create a multifunctional nanomotor with positional control of the catalyst, it would open up new opportunities to study motion, drug release, communication and interactions at the nanoscale.

For the design of biodegradable nanomotors it is crucial to consider its materials. To date, bulk of research has focused on creating artificial motors comprising metals (metal oxides) and/or non-degradable components and are therefore not relevant for biomedical applications[14–19]. Furthermore, to mimic natural motors, a controlled bottom up approach is necessary to create soft supramolecular motors. Our group has developed the first supramolecular nanomotors based on polymersomes[20,21]. Polymersomes are artificial bilayer vesicles made from amphiphilic block copolymers. The block copolymers self-assemble in aqueous solutions into spherical vesicles. Introducing osmotic pressure can induce shape transformation of the polymersomes into various morphologies[22]. The shape transformation depends on the reaction conditions, such as the polymer composition, polymer length and organic solvent(s)[23]. By controlling these conditions, different polymersome sizes and shapes can be made, including stomatocytes; bowl-shaped vesicles with an opening on one side[24,25]. Encapsulating a catalyst inside the stomatocyte cavity results in the formation of a nanomotor when fuel is present[26]. Besides the constituents of a nanomotor, also the fuel should be non-toxic, limiting the use to biocatalysts, such as enzymes. Lately, multiple examples of nanomotors utilizing enzymes have already been reported[27–30]. In our group a mild encapsulation technique for stomatocytes was developed to encapsulate enzymes by controlling the size of the opening with only small amounts of organic solvent[31]. This method yielded nanomotors at physiological levels of glucose, showing promising results for biomedical applications. However, this method requires precise control and tuning over the opening of the stomatocytes to prevent outflow of the enzymes. Furthermore, the catalyst encapsulation via this method is dependent on statistical encapsulation and therefore not well controlled. Finally, the stomatocytes were made from non-degradable poly(ethylene glycol)-b-polystyrene (PEG-PS), which should ideally be replaced with a degradable block copolymer. However, changing the polymer composition affects the response of the polymersome to the self-assembly, shape transformation, and the encapsulation method, thus complicating even more the control over the required parameters. Taken together, the encapsulation method requires time and effort and that hinders the reproducibility for a broad application range. Covalent binding of the enzymes inside the stomatocytes instead of statistical encapsulation would solve these problems, as covalent binding improves the efficiency and reproducibility of the enzyme incorporation. Furthermore, the size of the opening is less relevant, as it does not need to retain the enzymes inside the stomatocyte. This will open up the possibility for future mechanistic studies on the effect of the opening of the stomatocyte, catalyst loading and mechanism of motion.

A universal method was created to form nanomotors based on self-assembly of polymersomes as depicted in Fig. 1. The method is shortly described in this section, detailed information of each step and the characterization is discussed in the results section. We used biodegradable poly(ethylene glycol)-b-poly(D,L-lactide) (PEG-PDLLA) block copolymers, including an azide terminated polymer that was used as functional handle. Stomatocytes were formed containing 5 wt% azide handles on both the outside of the vesicle and the inner cavity. The azide handles on the outside were subsequently reduced into amine groups by immobilized tris(2-carboxyethyl)phosphine (TCEP) beads, effectively creating a second functional handle. The TCEP beads are larger than the opening of the stomatocytes, preventing the reduction of the azide handles in the inner cavity. Via 1-ethyl-3-(3-dimethylaminopropyl)carbodiimide (EDC) coupling any molecule containing a carboxylic acid can be coupled to the amine groups on the outer surface of the stomatocyte. Subsequently, two enzymes, catalase (Cat) and glucose oxidase (GOx) were covalently attached in the inner cavity of the stomatocyte. Both enzymes were first coupled to a dibenzocyclooctyne-sulfo-N-hydroxysuccinimidyl (DBCO-Sulfo-NHS) linker, providing the enzymes with a triple bond for strain-promoted azide-alkyne cycloaddition (SPAAC). The enzymes were then added to the stomatocyte sample, covalently binding to the azide groups in the inner cavity. Addition of physiological amounts of glucose solution activated the glucose oxidase to break down the glucose into gluconolactone and hydrogen peroxide. The formed hydrogen peroxide was then decomposed by the catalase into water and oxygen, which propelled the nanomotors forward. In the next section the detailed information and characterization for the PEG-PDLLA nanomotors is given.

## Results and Discussion

**Formation of PEG-PDLLA stomatocytes with azide handles.** Three different PEG-PDLLA polymers were synthesized using ring opening polymerization (ROP), including an azide functionalized polymer; mPEG$_{22}$-PDLLA$_{90}$, mPEG$_{44}$-PDLLA$_{90}$ and N$_3$-PEG$_{67}$-PDLLA$_{75}$. The product compositions were calculated from their respective NMR and GPC spectra (Supplementary Table 1 and Supplementary Fig. 1). The three polymers were mixed in the following weight ratio; 10:9:1 (for mPEG$_{22}$-PDLLA$_{90}$, mPEG$_{44}$-PDLLA$_{90}$, and N$_3$-PEG$_{67}$-PDLLA$_{75}$, respectively) and dissolved in organic solvent (1:4 tetrahydrofuran (THF) to dioxane). An equivalent of water was added slowly at 1 mL/h, inducing self-assembly into spherical polymersomes. After dialysis against 10 mM NaCl, stomatocyte morphologies with open necks were obtained, as shown by cryo-transmission electron microscopy (cryo-TEM) (Fig. 2). The opening of the stomatocytes was measured from the cryo-TEM images to be around 30 nm (Supplementary Fig. 2). Both cryo-TEM images and Dynamic Light Scattering (DLS) measurements showed an average size of around 370 nm (Supplementary Figs. 2 and 3).

**Reduction of azide handles on the outside of the stomatocyte.** The azide handles on the outside of the stomatocytes were reduced by commercially available Pierce immobilized TCEP resin. DLS measurements show an average particle size of ~10 μm (PDI of 0.194), confirming that the TCEP beads cannot enter the stomatocytes cavity. The reduction was quantified by coupling a dye to the azide groups and comparing the fluorescence intensity before and after reduction. In total, the fluorescence signals of 4 different samples were measured in triplicate (Fig. 2c). The first sample consisted of stomatocytes without reduction, still containing all azide groups for binding. The second group was

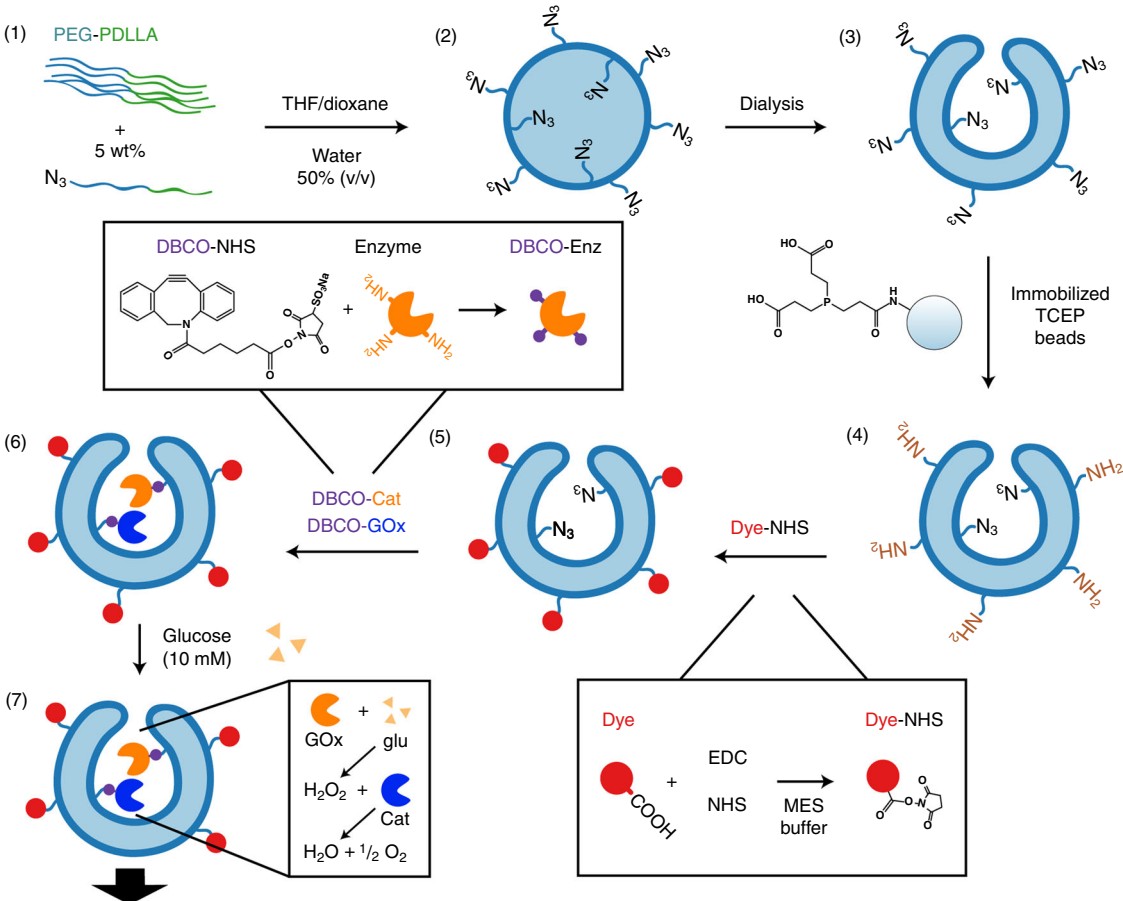

**Fig. 1** Universal method to form polymeric nanomotors with spatial control over catalyst. (1) Formation of amphiphilic poly(ethylene glycol)-b-poly(D,L-lactide) (PEG-PDLLA) block copolymers, including an azide terminated block copolymer to act as functional handle. (2) Self-assembly of block copolymers with 5 wt% azide polymer into spherical polymersomes. (3) Shape transformation of the polymersomes into stomatocytes. Azide handles in the inner membrane of the stomatocyte are not shown for clarity. (4) Reduction of azide groups to amine groups on the outer surface by immobilized TCEP beads that are too large to enter the cavity. (5) Attachment of a dye to the amine groups on the outside via EDC coupling. (6) Covalent coupling of enzymes into the stomatocyte cavity via SPAAC. (7) Addition of physiological amounts of glucose activates the enzyme cascade to produce water and oxygen that propel the system forward.

treated with the TCEP beads and was expected to contain around half of the azide groups. The third group was reduced with free TCEP, which would theoretically lead to complete reduction of the azide groups, as the TCEP is able to enter the stomatocyte cavity. The final sample consisted of only PEG-PDLLA polymer without azide groups as negative control. An Alexa Fluor 488-alkyne dye was coupled via copper(I)-catalyzed alkyne-azide cycloaddition (CuAAC) to the available azide groups for each sample. Afterwards, all samples were dissolved in THF to destroy the vesicle structures. The fluorescence signal of each sample was measured and is depicted in Fig. 2c. From the peak signals at 540 nm was calculated that around 45% azide groups remained after the reduction with the TCEP beads, indicating that only the outside azide groups were reduced. The reduction with free TCEP resulted in a fluorescence intensity of 8%, suggesting that most of the azide groups were reduced.

To confirm spatial control over the reduction and thereby the localization of the catalyst, gold nanoparticles (20 nm) with NHS linkers were coupled to the reduced stomatocytes and visualized with cryo-TEM. The gold nanoparticles coupled to the outside of reduced stomatocytes, showing partial reduction, as visualized in Fig. 2b. The control experiment with non-reduced stomatocytes did not show any binding (Supplementary Fig. 4). Important to note is that any molecule containing a carboxylic acid group could be attached, such as receptors, peptides, or sugar groups, depending on the desired application. Therefore, to proof this

concept, an Alexa Fluor 647-NHS dye was attached to the amine groups on the outer surface of reduced stomatocytes via EDC coupling. With confocal microscopy the attachment of the dye was visualized (Supplementary Fig. 5).

**Coupling enzymes inside stomatocytes to form nanomotors.** Finally, enzymes were covalently coupled inside the stomatocyte cavity. Two different samples were made, one containing the enzyme catalase (Cat) and the other one containing two enzymes, catalase (Cat) and glucose oxidase (GOx). For both enzyme systems, the enzymes were coupled to DBCO-NHS, which was visualized with UV–Vis measurements, showing additional absorbance peaks at 310 nm, specific for the DBCO molecule (Supplementary Fig. 6a). A ratio of ten linker molecules per enzyme was added, which was verified by the ultraviolet (UV)–visible (Vis) experiments (Supplementary Fig. 6b). The activity of the enzymes after binding with DBCO-NHS was tested via an Amplex red assay. Two ratios of 10:1 and 30:1 DBCO linker molecules per catalase were tested and compared to unmodified catalase, they retained 98.9 and 96.2% activity, respectively, showing no significant loss in activity after binding (Supplementary Fig. 7). After washing the enzymes to remove the non-reacted DBCO linker, they were added to the stomatocyte sample for the coupling. The coupling of the enzymes to the azide functionalized polymer was visualized with sodium dodecyl

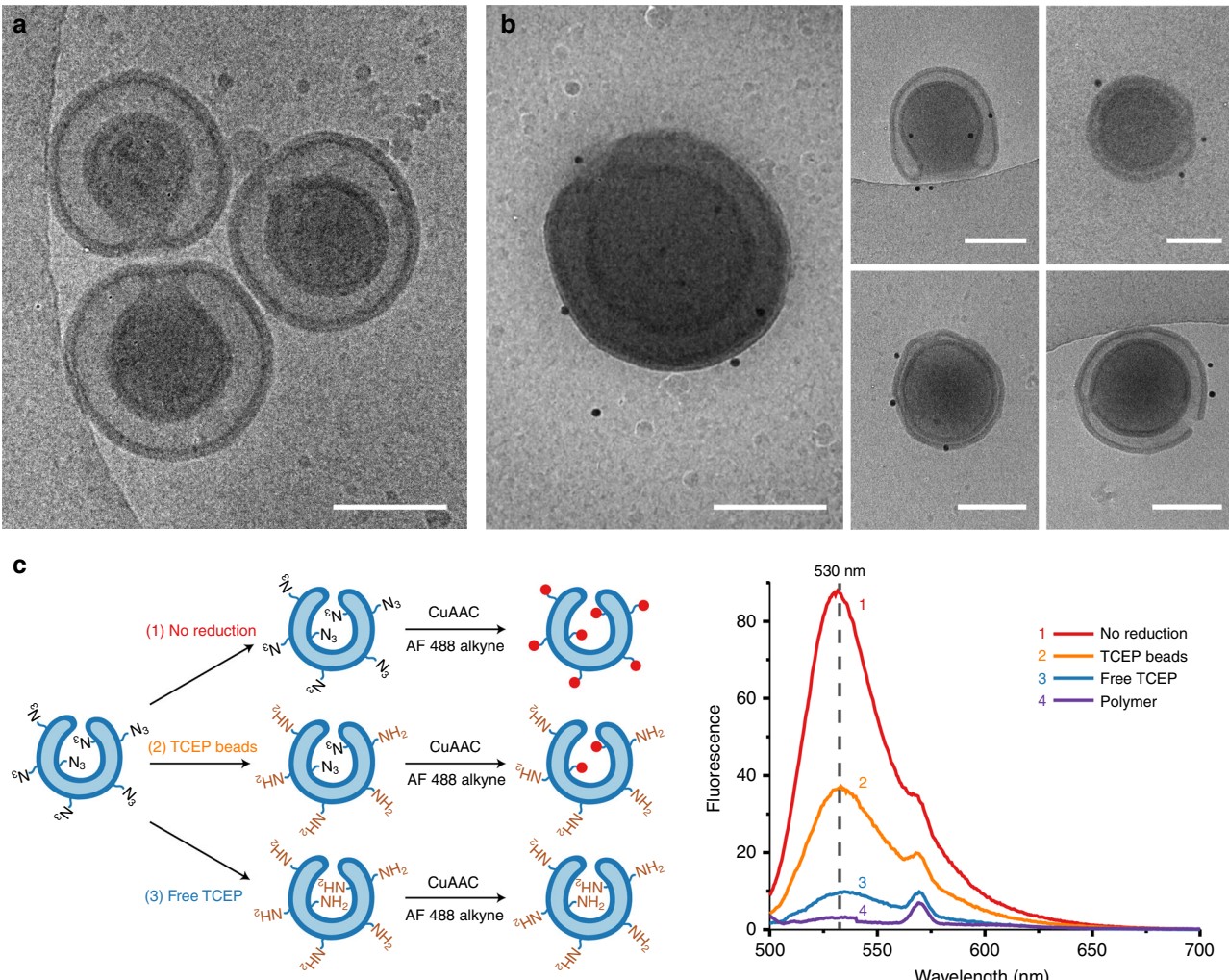

**Fig. 2** Characterization of PEG-PDLLA stomatocytes with azide handles. **a** Cryo-TEM picture of PEG-PDLLA stomatocytes. Scale bar, 250 nm. **b** Cryo-TEM pictures of PEG-PDLLA stomatocytes after reduction with TCEP beads and subsequent reaction with NHS functionalized gold nanoparticles (20 nm), showing binding to the outer surface. Scale bars, 250 nm. **c** Reduction assay; fluorescence spectra of Alexa Fluor 488-alkyne clicked to available azide groups before and after reduction. 1 (red line) Fluorescence spectrum without reduction, showing maximum fluorescence signal. 2 (orange line) Fluorescence after reduction with TCEP beads, showing 45% signal, indicating approximately half the azide groups were reduced. 3 (blue line) Fluorescence after reduction with free TCEP, showing 8% signal, showing almost complete reduction. 4 (purple line) Fluorescence signal of polymer only as negative control.

sulfate-polyacrylamide gel electrophoresis (SDS-PAGE) (Fig. 3a). For this experiment, catalase functionalized stomatocytes were dissolved in dioxane to precipitate the coupled enzymes and to remove the unreacted block copolymers. The cat-polymer conjugates were isolated and run on a 4–20% gel, together with a catalase concentration series to estimate the enzyme loading. The gel showed the bands around 60 kDa corresponding to the catalase subunit. For the conjugate sample an extra band was expected around 70 kDa, due to the attachment of the polymer. As catalase consists of 4 subunits, a ratio of around 3:1 in intensity was expected for the bands corresponding to the unbound subunits and the enzyme-polymer conjugates, respectively, which was visible on the gel (Fig. 3a). After formation of the nanomotors, their movement speed was tested in different amounts of fuel. The motors containing both GOx and Cat were tested with fuel concentrations of 10 and 30 mM glucose, whereas the motors containing only Cat were tested at 5 and 10 mM $H_2O_2$ (Fig. 3b). For each sample videos of 60 s were recorded and analyzed using Nanosight NTA2.2 software (Video S1-3). After tracking, the $x$ and $y$ coordinates of at least 100 particles per group were used to calculate their mean-squared displacement

(MSD). These MSD values were plotted against $\Delta t$, showing increased movement speeds at higher fuel concentrations (Fig. 3b). From their curves, the average speeds were calculated and are depicted in Fig. 3c). The motion of enzyme functionalized stomatocytes show a parabolic fit, typical for bubble propulsion. Recently, our group has shown that bowl-shaped micromotors can help in confinement and nucleation of the bubbles and play a templating role for the formation of the bubbles, which collapse at regular times due to the filling of the cavity[32]. We think in the same way, oxygen nanobubbles are formed inside of the cavity and the regular collapse is responsible for the ballistic motion observed. In a control experiment, the motion of the enzyme loaded stomatocytes was compared with enzyme functionalized spherical polymersomes. First, spherical polymersomes with 5 wt % azide handles were created by dialysis against water instead of salt (Supplementary Fig. 8). To these polymersomes, catalase with the DBCO linker was added resulting in the random attachment of catalase on the spherical surface. The movement speed was tested at different fuel concentrations and the spherical polymersomes showed also increased movement speed, but with a linear fit, indicating enhanced diffusion (Supplementary Fig. 9).

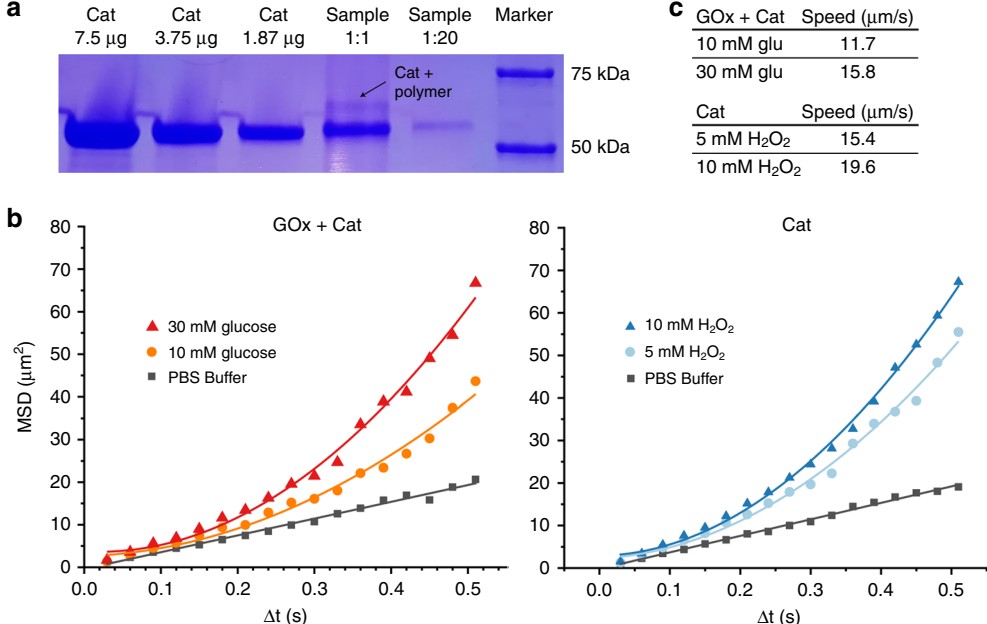

**Fig. 3** Characterization of PEG-PDLLA nanomotors. **a** SDS-PAGE demonstrating successful coupling of catalase to the polymer, indicated by black arrow. **b** Mean-squared displacement (MSD) values of stomatocytes obtained with various fuel concentrations, 30 mM glucose (red triangles), 10 mM glucose (orange circles), PBS buffer (gray squares), 10 mM $H_2O_2$ (darker blue triangles), 5 mM $H_2O_2$ (lighter blue circles). **c** Overview of movement speeds of the nanomotors at various fuel concentrations for both enzyme combinations.

This behavior is explained for the random attachment of catalase on the surface. This is comparable to previous studies with attachment of catalase to rod shaped nanomotors[20].

In conclusion, we have shown the formation of a biodegradable supramolecular nanomotor with spatial control over the catalyst attachment and with a multivalent design. Well-defined stomatocyte structures were generated with the biodegradable PEG-PDLLA block copolymer, containing functional azide handles. Selective reduction on the outside surface of the stomatocytes with TCEP immobilized beads provided the structures with a secondary handle for coupling another functional group. Attachment of fluorescent dyes and functionalized gold nanoparticles showed the selective reduction of the outer azide groups, allowing the binding of enzymes inside the stomatocytes via click-reactions. SDS-PAGE and NTA experiments proved the formation of the nanomotor, which showed an increase in MSD after the addition of fuel.

## Methods

**Synthesis of poly(ethylene glycol)-b-poly(D,L-lactide)**. Poly(ethylene glycol)-b-poly(D,L-lactide) (PEG-PDLLA) was synthesized by ring opening polymerization (ROP). For the formation of $PEG_{22}$-$PDLLA_{90}$, 0.2 mmol methoxy-PEG-OH macroinitiator (194 mg) was mixed with 18 mmol D,L-Lactide (2.6 g, 13 wt% PEG in total). For the other polymer compositions the amounts were adjusted to obtain $PEG_{44}$-$PDLLA_{90}$ (397 mg methoxy-PEG-OH, 2.6 g D,L-Lactide) and $N_3$-$PEG_{67}$-$PDLLA_{75}$ (150 mg $N_3$-PEG-OH, 634 mg D,L-Lactide). First, the reagents were dried by adding dry toluene and removing the solvent under reduced pressure. Then, 15 mL dry DCM with 0.1 mmol DBU (15 µL, 0.1 mmol) was added to the dried material under argon. The reaction was left to proceed for 3–4 h at 30 °C. After finishing the polymerization, the mixture was washed twice with 1 M $KHSO_4$, dried with $Na_2SO_4$ and filtered off. The polymer was concentrated by evaporating most solvent (~4 mL) and then precipitated in ice cold diethyl ether (100 mL). The waxy substance was partly dried under nitrogen, dissolved in 1,4 dioxane (5 mL) and lyophilized to yield a white powder (~80% yield). Polymerization was checked with NMR and GPC (Supplementary Table 1 and Supplementary Fig. 1).

**Preparation of PEG-PDLLA stomatocytes with 5% azide handles**. In total 10 mg PEG-PDLLA polymer was weighed in a glass vial with stirring bar, 0.5 mg $N_3$-$PEG_{67}$-$PDLLA_{75}$, 4.5 mg $PEG_{22}$-$PDLLA_{90}$, and 5 mg $PEG_{44}$-$PDLLA_{90}$. The mixture was dissolved in 1 mL of organic solvent (1:4, THF:dioxane v/v). The vial was closed with a rubber septum and the mixture was stirred for 30 min at 800 rpm. Subsequently, 1 mL Milli-Q water (50 wt%) was added via a syringe pump at 1 mL per hour until a

cloudy suspension was obtained. The suspension was transferred to a pre-hydrated membrane (Spectra/Por, molecular weight cutoff: 12–14 kDa) and dialyzed against 1 L of 10 mM NaCl for 24 h in a fridge at 4 °C, with a solution change after 1 h to obtain the stomatocyte shape. Samples were stored in the fridge at 4 °C.

**Reduction of the outer azide handles**. For reduction of only outer azide handles: 200 µL of immobilized tris(2-carboxyethyl)phosphine (TCEP) immobilized on silica beads was centrifuged at 106 rcf to remove all liquid. To the beads, a suspension of azide functionalized stomatocyte vesicles (200 µL of 5.0 mg/mL polymersomes) was added. The reaction mixture was stirred at 100 rpm for 36 h at room temperature, with halfway a change with fresh TCEP beads. After reduction, the stomatocyte solution was diluted with Milli-Q to 3 mL and centrifuged two times for 1 min at 106 rcf to remove the TCEP beads. The supernatant was concentrated by centrifuging for 15 min at 2655 rcf and dissolved up to 5.0 mg/mL polymersomes in $NaHCO_3$ buffer (0.1 M, pH = 8.3).

**Attachment of Alexa Fluor 647-NHS to amide handles**. In all, 0.02 mg of Alexa Fluor 647-NHS ester was dissolved in 3 µL DMSO and added to 200 µl reduced stomatocytes, the reaction mixture was left stirring for 16 h in the dark at room temperature. The unreacted compounds were removed by spin filtration with 0.22 µm spin filters at 2655 rcf for 10 min. The binding was visualized with confocal and with fluorescence measurements.

**Attachment of gold nanoparticles to amide handles**. The conjugation protocol was performed as described in the assay provided in the kit. The stomatocytes were first concentrated by centrifuging for 15 min at 2655 rcf and the supernatant was removed. Afterwards, the stomatocytes were resuspended in reaction buffer and mixed with the NHS functionalized gold nanoparticles. The solution was mixed on the roller bench for 24 h at room temperature. Stomatocytes were collected by centrifugation and washing several times for 15 min at 2655 rcf.

**Coupling of DBCO-NHS to enzymes**. The enzyme of choice is dissolved in 500 µL PBS buffer (catalase 2 mg, $8.3 \times 10^{-6}$ mmol glucose oxidase 6 mg, $3.3 \times 10^{-5}$ mmol). To each enzyme solution a tenfold excess of DBCO-Sulfo-NHS was added and mixed for 24 h. Binding was verified by measuring absorbance using a UV–Vis spectrophotometer. The data and calculations are available in the Source Data file.

**Coupling of enzymes inside the stomatocyte cavity**. The enzyme samples were mixed with the stomatocyte samples in a 5 mL glass vial with stirring bar and mixed for 24 h at 100 rpm. The sample was centrifuged and washed twice for 15 min at 2655 rcf with PBS buffer to remove unbound enzyme. Finally, the sample was washed with PBS buffer in 0.22 µm spin filters at 2655 rcf for 10 min. The nanomotors were collected in 100–200 µL PBS buffer and stored in the fridge at 4 °C.

**Quantification of azide reduction**. The reduction was quantified by measuring fluorescence intensity after binding an Alexa Fluor 488-alkyne with click reaction. To different samples (non-reduced, TCEP beads reduced and free TCEP reduced) 0.05 mg of Alexa Fluor 488-Alkyne, 0.25 mg of $CuSO_4 \cdot 5H_2O$ and 0.16 mg of (+)-Sodium L-ascorbate were added and stirred for 12 h in the dark at room temperature. Unreacted compounds were removed by spin filtration with 0.22 μm spin filters at 2655 rcf for at 10 min. The stomatocytes were dissolved in 2 mL THF, so that the vesicles were destroyed completely. Fluorescence intensities were measured with a Spectrofluorometer.

**Sodium dodecyl sulfate-polyacrylamide gel electrophoresis**. Stomatocyte motors with catalase attached (400 μL) were centrifuged at 10,621 rcf for 10 min for collection. The supernatant was discarded and the motors were dissolved in 1 mL of dioxane to dissolve the polymers. The sample was again centrifuged and washed twice with dioxane to collect the clustered enzymes for 2 min at 10,621 rcf. Finally, the enzymes were dissolved in 50 μL Milli-Q water. The sample and a 20x dilution were loaded in a 4–20% SDS-precast protein gel (Bio-Rad), together with several concentrations of catalase (7.5, 3.75 and 1.87 μg) and the Precision Plus Unstained marker. The unprocessed scan is supplied in the Source Data file.

**Reporting summary**. Further information on research design is available in the Nature Research Reporting Summary linked to this article.

## Data availability

The experiment data that support the findings of this study are available from the corresponding author upon reasonable request. The source data underlying Figs. 2c, 3a–c and Supplementary Figs. 1b, 3a, c, 6a, b, 7, 9 are provided as a Source Data file.

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

## Acknowledgements

We acknowledge the NWO Chemische Wetenschappen VIDI Grant 723.015.001 for financial support. We also acknowledge support from the Ministry of Education, Culture and Science (Gravitation program 024.001.035).

## Author contributions

J.T. and D.A.W. conceived and designed the experiments. J.T. and F.C. performed the experiments.

## Competing interests

The authors declare no competing interests.
