## [Peer Review File · Nature Communications]

Reviewers' comments:

Reviewer #1 (Remarks to the Author):

The authors prepared reactive stomatocytes using a self-assembly procedure that has been reported by the authors earlier in various publications. These stomatocytes have been used as nanomotors in earlier work, but here the authors tried to place catalysts with high control inside of the stomatocytes only. This was achieved by selective reduction using beads, which I think is a really elegant approach. As a result, bowl-shaped structures were obtained that contained enzymes on the interior. These enzymes were able to undergo an enzymatic reaction, which produces gas. As a result, a nanorocket was obtained. Overall, an interesting approach, but the manuscript needs additional control experiments, which are outlined below

1. The authors claim that it is the size of the beads that prevents entry into the centre. As this can probably not be proven directly, the authors have used a control experiment showing that the amino groups are only localized on the outside by reaction of these functional groups with gold nanoparticles. While this is an elegant method, I am not too sure that Figure 2B provides sufficient proof. There are some gold nanoparticles attached, but I can also see some free particles and some on the inside. The authors only showed one stomatocyte here and it would be nice to see an overview of many to confirm that this is not a singled out TEM.

2. The authors confirmed the successful reaction between the enzyme and the DBCO-NHS linker using UV-Vis. This technique could probably be used to quantify the amount of linkers per enzyme. Mass spec could also be a good technique. Does the linker affect the enzymatic activity?

3. How many enzymes can be found in stomatocytes? Has the enzymatic activity been affected by binding to the surface?

4. What do these nanoparticles look like after enzyme attachment? The authors need to provide structural analysis to ensure that enzyme conjugation did not affect the shape of the nanoparticle

4. It is clear that the enzymes on the surface nicely increase the movement of the particles. Similar to the authors' earlier work, fuel was generated on the inside resulting in a nanorocket. What is not clear is if this movement is unique to these stomatocytes? What would happen to the speed if the enzymes are attached to polymersomes? And what about enzymes that are attached to the inside AND outside of the stomatocytes? I strongly recommend carrying out a few more control experiments here. I am also thinking that the faster movement could be simply the presence of oxygen bubbles in the solution that push the nanoparticles around.

Reviewer #2 (Remarks to the Author):

In this exciting manuscript, biodegradable nanomotors are realized which have been selectively modified on the inner part of the vesicles and the outer shell. For the first time, biodegradable PEG-PDLLA block copolymer has been successfully used to prepare well-defined stomatocytes (370 nm) with a cavity in the block copolymer membrane of about 30 nm. In that the authors made use of their intensive experience in defined polymer synthesis and preparation of well-defined assemblies. The most innovative and exciting feature of this study is the approach to be able to obtain special control of reactive functions. Here, the authors used a reagent which was immobilized on silica beads, tris(2-carboxyethyl)phosphine, to reduce selectively the azides on the outside of the vesicle. The beads had been larger than the cavity in the stomatocytes, and thus could not reach the azide functions inside the vesicle. This selectivity was very well verified in various control experiments. This approach to allow special control of specific reactions, is highly interesting and of broad importance in a wide area of application!

In the present case, the author used the non-reduced azide functions in the inner lumen of the vesicles for covalent immobilization of enzymes which produce the oxygen fuel from glucose for moving the nanomotors. The outer amine functions were used for fluorescence labeling, but of course, various moieties can be attached to the outer wall of the stomatocytes using this approach, exemplified also by the authors with gold nanoparticles.

The authors characterized their nanomotors very well and quantified the special selective functionalization. Also, the nanomotors were nicely characterized with regard to displacement speed, also depending on the supplied glucose concentration. Looking at the videos, however, one can see that it is not possible to control well the directionality of the movement of the motors. In my opinion, the manuscript is suited for Nature Communication since it broadens significantly not only the field of functional nanomotors, but presents a universal concept for spatial controlling reactions.

There are only minor revisions needed. The most important one is that the authors have to provide the exact size and size distribution of their immobilized TCEP beads (not just making the remarks that the beads are larger than the cavity). There is also a small inconsistency: in the experimental part the author call it: "Pierce immobilized TCEP disulfide-reducing resin", whereas in the text it is only mentioned as "immobilized tris(2-carboxyethyl)phosphine (TCEP) beads", not indicating any disulfides. In general it would be good to get also information on the loading with TCEP on the particles and more on the support silica particles.

Similarly, the authors have to give the details on their used gold nanoparticles (size and functionality? information missing in SI). Also details on the source (and maybe activity data) of the enzymes are missing.

One small point: it would be good to explain in the figure caption Figure 3 MSD (mean squared displacement).

Dear Prof. Dr Johannes Kreuzer,

Thank you very much for the referee report for our manuscript "Spatial control over catalyst positioning on biodegradable polymeric nanomotors" by B. Jelle Toebes, F. Cao and Daniela A. Wilson. We appreciate the comments and suggestions of both referees and we have revised the manuscript accordingly. Our answer to the questions by the referees and the revision that has been done is stated below. We hope that with these additions, the manuscript is now acceptable for publication.

Reviewer #1 (Remarks to the Author):

The authors prepared reactive stomatocytes using a self-assembly procedure that has been reported by the authors earlier in various publications. These stomatocytes have been used as nanomotors in earlier work, but here the authors tried to place catalysts with high control inside of the stomatocytes only. This was achieved by selective reduction using beads, which I think is a really elegant approach. As a result, bowl-shaped structures were obtained that contained enzymes on the interior. These enzymes were able to undergo an enzymatic reaction, which produces gas. As a result, a nanorocket was obtained. Overall, an interesting approach, but the manuscript needs additional control experiments, which are outlined below.

We are pleased with the positive evaluation of referee 1 and we would like to thank him/her for the suggestions and effort to improve the manuscript. Here are our answers to his/her comments, point by point in blue:

1) The authors claim that it is the size of the beads that prevents entry into the centre. As this can probably not be proven directly, the authors have used a control experiment showing that the amino groups are only localized on the outside by reaction of these functional groups with gold nanoparticles. While this is an elegant method, I am not too sure that Figure 2B provides sufficient proof. There are some gold nanoparticles attached, but I can also see some free particles and some on the inside. The authors only showed one stomatocyte here and it would be nice to see an overview of many to confirm that this is not a singled out TEM.

Answer: We would like to thank this reviewer for his/her suggestion to clarify the experiments performed with gold nanoparticles. We agree that this single TEM picture is not showing enough evidence of binding. Therefore we have included several other pictures of the stomatocytes after binding with the gold nanoparticles. All the images provided are cryo-TEM images since TEM images are not possible for the polymer of choice due to collapse of the vesicles caused by drying effects or strong vacuum. It is therefore quite difficult to be able to capture in the same image multiple stomatocytes clustered together in a cryo-TEM technique however we can provide multiple structures that have been observed within the grid. The additional pictures showed the particles attached to the wall of the nanomotor. Yet, it is important to note the particles are able to bind all over the stomatocyte exterior surface, and since TEM and cryo-TEM technique is a 2D technique the projected image would contain also particles that seem to be in the inside. Therefore with this technique it is difficult to distinguish between particles that are accidentally captured inside or bound outside of the stomatocytes in the center. Nevertheless based on the intensity of the particles they seem to be on the outside. The other difficulty of the direct method of choice is that the coupling reaction occurs between two particles therefore it is more challenging for the reaction to take place with high density on the surface compared to a small molecule such as a fluorophore. Nevertheless with the two direct and indirect proofs as well as the additional functioning of the nanomotors we are confident to conclude that the reaction is selective only to the outside surface.

2) The authors confirmed the successful reaction between the enzyme and the DBCO-NHS linker using UV-Vis. This technique could probably be used to quantify the amount of linkers per enzyme. Mass spec could also be a good technique. Does the linker affect the enzymatic activity?

Answer: We would like to thank this reviewer for the feedback. Indeed it is important to consider the enzymatic activity after binding with the linker. We have performed an enzyme activity assay with various enzyme to linker ratios. The enzyme activity was not significantly reduced up to 30 linker molecules per enzyme. For our experiments we have used a fivefold excess of linker molecules, which we expect to be all bound to the enzyme, as catalase has many sites for coupling.

3) How many enzymes can be found stomatocytes? Has the enzymatic activity been affected by binding to the surface?

Answer: We would like to thank the referee for this comment. Indeed this is a very interested question and a question that we have asked ourselves. However, based on the techniques we have in hand we were not able to assess quantitatively the amount of enzymes per stomatocytes. This is partially due to the fact the stomatocytes are obtained from a mixture of multiple block copolymers (with linkers and without linkers), the remaining accessible linkers can further interact with the enzyme after the disassembly of the structure and difficulty in separation and quantification of the polymer-enzyme hybrid system. However based on the activity of the enzyme and the resulted thrust it seems that the attachment did not affect much the activity.

4) It is clear that the enzymes on the surface nicely increase the movement of the particles. Similar to the authors earlier work, fuel was generated on the inside resulting in a nanorocket. What is not clear if this movement is unique to these stomatocytes? What would happen to the speed if the enzymes are attached to polymersomes? And what about enzymes that are attached to the inside AND outside of the stomatocytes? I strongly recommend carrying out a few more control experiments here. I am also thinking that the faster movement could be simply the presence of oxygen bubbles in the solution that push the nanoparticles around.

Answer: We thank this referee for these suggestions. We have performed some movement studies with catalase bound to spherical polymersomes as a control. These polymersomes showed enhanced diffusion in present of fuel, much like earlier reported nanotubes (Polymer Chem., 2018, 9(23), 3190-3194). Indeed we think the movement is specific to the stomatocyte shape and it is not related to the presence of bubbles in solution. In our recent manuscript Adv Funct Mater 2019 we have demonstrated that the bowl shape micromotors can help in confining and nucleation of the bubbles and play the role of template for the formation of the bubbles, which collapse at regular times due to the filling of the cavity. We think in the same way, oxygen nanobubbles are formed inside of the cavity and the regular collapse is responsible for the ballistic motion observed. Although we cannot prove the formation and collapse of the oxygen nanobubbles since are too small to be observed with our current techniques the high speeds observed correlate well with bubble propelled systems. The control experiments with vesicles attached to the surface of polymersomes showed only enhanced motion confirming our previous observation that in this case the mechanism of motion is driven by self-diffusiophoretic mechanism as observed in tubular polymersomes and is not due to bubbles formed outside in solution.

Reviewer #2 (Remarks to the Author):

In this exciting manuscript, biodegradable nanomotors are realized which have been selectively modified on the inner part of the vesicles and the outer shell. For the first time, biodegradable PEG-PDLLA block copolymer have been successfully used to prepare well defined stomatocytes (370 nm)

with a cavity in the block copolymer membrane of about 30nm. In that the authors made use their intensive experience in defined polymer synthesis and preparation of well-defined assemblies. The most innovative and exciting feature of this study is the approach to be able to obtain special control of reactive functions. Here, the authors used a reagent which was immobilized on silica beads, tris(2-carboxyethyl)phosphine, to reduce selectively the azides on the outside of the vesicle. The beads had been larger than the cavity in the stomatocytes, and thus could not reach the azide functions inside the vesicle. This selectivity was very well verified in various control experiments. This approach to allow special control of specific reactions, is highly interested and of broad importance in a wide area of application! In the present case, the author used the non-reduced azide functions in the inner lumen of the vesicles for covalent immobilization of enzymes which produce the oxygen fuel from glucose for moving the nanomotors. The outer amine functions were used for fluorescence labeling, but of course, various moieties can attached to the outer wall of the stomatocytes using this approach, exemplified also by the authors with gold nanoparticles. The authors characterized their nanomotors very well and quantified the special selective functionalization. Also, the nanomotors were nicely characterized with regard to displacement speed, also depending on the supplied glucose concentration. Looking at the videos, however, one can see that it is not possible to control well the directionality of the movement of the motors. In my opinion, the manuscript is suited for Nature Communication since it broadens significantly not only the field of functional nanomotors, but presents a universal concept for spatial controlling reactions. There are only minor revisions needed.

We are would like to thank referee 2 for his/her kind words and comments to improve the manuscript. Here are our answers to his comments, point by point in blue:

1. The most important one is that the authors have to provide the exact size and size distribution of their immobilized TCEP beads (not just making the remarks that the beads are larger than the cavity). There is also a small inconsistency: in the experimental part the author call it: "Pierce immobilized TCEP disulfide-reducing resin", whereas in the text it is only mentioned as "immobilized tris(2-carboxyethyl)phosphine (TCEP) beads", not indicating any disulfides. In general it would be good to get also information on the loading with TCEP on the particles and more on the support silica particles. Similarly, the authors have to give the details on their used gold nanoparticles (size and functionality? information missing in SI).

Answer: We thank the referee for these suggestions. We have measured the size and size distribution of the immobilized TCEP beads and included these in the supporting information. Furthermore, we have updated the supporting information with additional information and details of the TCEP beads as well as the gold nanoparticles.

2. Also details on the source (and maybe activity data) of the enzymes are missing.

Answer: We would like to thank this reviewer for this comment. We have performed an enzyme activity assay with various enzyme to linker ratios to ensure that the activity is not affected by binding and included these in the supporting information and furthermore updated the details in the supporting information.

3. One small point: it would be good to explain in the figure caption Figure 3 MSD (mean squared displacement).

Answer: We thank this reviewer for the helpful feedback. We have updated the caption in Figure 3 to explain the MSD.

We hope that with these modifications the manuscript is now acceptable for publication in Nature Communications.

Sincerely yours, Daniela Wilson

REVIEWERS' COMMENTS:

Reviewer #1 (Remarks to the Author):

The authors have addressed all my comments, apart from one:

The authors confirmed the successful reaction between the enzyme and the DBCO-NHS linker using UV-Vis. This technique could probably be used to quantify the amount of linkers per enzyme.

The authors provided me with a very useful answer on the importance of the bowl-shape. It would have been nice to discuss this a bit more in the manuscript

Reviewer #2 (Remarks to the Author):

The authors submitted an extensive rebuttal letter where they attend very well to all reviewer questions and remarks. In addition, they revised the manuscript accordingly. Significant new experiment had been carried out and added (to the main manuscript as well as to SI) to support the results and discussion which improved the manuscript substantially. E.g. additional cryo-TEM pictures are shown for the attachment of the gold nanoparticle to the surface of the, additional information on starting materials and experiments have been added to SI (including calibration curves and reference experiments), additional discussion and details have been added in the manuscript.

All open questions were fully answered, any needed corrections have been made and thus, the manuscript can now be recommended for publication in the present state.

REVIEWERS' COMMENTS:

Reviewer #1 (Remarks to the Author):

The authors have addressed all my comments, apart from one:

The authors confirmed the successful reaction between the enzyme and the DBCO-NHS linker using UV-Vis. This technique could probably be used to quantify the amount of linkers per enzyme.

We thank the reviewer to point out this experiment. We have performed the above-mentioned experiment and implemented this in the manuscript and Supplementary Information. We could indeed verify the amount of linkers per enzyme.

The authors provided me with a very useful answer on the importance of the bowl-shape. It would have been nice to discuss this a bit more in the manuscript.

This is also a useful comment. We have updated the manuscript to explain the importance of the bowl-shape on the movement of the nanomotors.

We hope that with these modifications the manuscript is now ready for publication in Nature Communications.